# Relationships between electrolyte and amino acid compositions in sweat during exercise suggest a role for amino acids and K$^+$ in reabsorption of Na$^+$ and Cl$^-$ from sweat

**Grace R. Murphy**[☺], **R. Hugh Dunstan**[iD]*[☺], **Margaret M. Macdonald**[☺], **Nattai Borges**[☺], **Zoe Radford**[‡], **Diane L. Sparkes**[‡], **Benjamin J. Dascombe**[‡], **Timothy K. Roberts**[‡]

University of Newcastle, Callaghan, NSW, Australia

☺ These authors contributed equally to this work.
‡ These authors also contributed equally to this work.
* hugh.dunstan@newcastle.edu.au

**Data Availability Statement:** All relevant data are within the manuscript.

## Abstract

Concentrations of free amino acids and [K$^+$] in human sweat can be many times higher than in plasma. Conversely, [Na$^+$] and [Cl$^-$] in sweat are hypotonic to plasma. It was hypothesised that the amino acids and K$^+$ were directly or indirectly associated with the resorption of Na$^+$ and Cl$^-$ in the sweat duct. The implication would be that, as resources of these components became limiting during prolonged exercise then the capacity to resorb [Na$^+$] and [Cl$^-$] would diminish, resulting in progressively higher levels in sweat. If this were the case, then [Na$^+$] and [Cl$^-$] in sweat would have inverse relationships with [K$^+$] and the amino acids during exercise. Forearm sweat was collected from 11 recreational athletes at regular intervals during a prolonged period of cycling exercise after 15, 25, 35, 45, 55 and 65 minutes. The subjects also provided passive sweat samples via 15 minutes of thermal stimulation. The sweat samples were analysed for concentrations of amino acids, Na$^+$, Cl$^-$, K$^+$, Mg$^{2+}$ and Ca$^{2+}$. The exercise sweat had a total amino acid concentration of 6.4 ± 1.2mM after 15 minutes which was lower than the passive sweat concentration at 11.6 ± 0.8mM ($p<0.05$) and showed an altered array of electrolytes, indicating that exercise stimulated a change in sweat composition. During the exercise period, [Na$^+$] in sweat increased from 23.3 ± 3.0mM to 34.6 ± 2.4mM ($p<0.01$) over 65 minutes whilst the total concentrations of amino acids in sweat decreased from 6.4 ± 1.2mM to 3.6 ± 0.5mM. [Na$^+$] showed significant negative correlations with the concentrations of total amino acids ($r = -0.97$, $p<0.05$), K$^+$ ($r = -0.93$, $p<0.05$) and Ca$^{2+}$ ($r = -0.83$, $p<0.05$) in sweat. The results supported the hypothesis that amino acids and K$^+$, as well as Ca$^{2+}$, were associated with resorption of Na$^+$ and Cl$^-$.

## Introduction

It has been well documented that eccrine sweat produced for evaporative cooling in humans contains water, electrolytes, lactate, ammonia and urea [1–3]. The electrolytes predominantly

**Funding:** This work was supported by the Gideon Lang Research Foundation grant number G1701109 to RHD. The funders had no role in study design, data collection and analysis, decision to publish, or preparation of the manuscript.

**Competing interests:** The authors have declared that no competing interests exist.

consist of $Na^+$ and $Cl^-$ and to a lesser extent, $K^+$, $Mg^+$ and $Ca^+$. Sweat also contains an array of free amino acids including serine, histidine, ornithine, alanine, glycine and lysine that are present at concentrations several times higher than their corresponding levels in plasma [4–6]. Little is known about the function of free amino acids in sweat or why they do not mirror the relative plasma compositions. It has been proposed that at least some of the relative increases of certain amino acids such as serine, histidine and glutamic acid relates to their function as humectants in the natural moisturising factor (NMF) of the outer layers of the skin [5, 7, 8]. However, small contributions from the NMF would be unlikely to account for the >ten-fold increases in concentration observed in sweat relative to plasma[7].

Estimates of the number of eccrine sweat glands on the human body range between 1.6–4 x $10^6$, with the highest densities located on the palms, finger print ridges and the soles of the feet [9]. It has previously been thought that the primary source of sweat electrolytes was derived from blood plasma filtrate of serous secretion through epithelia that have low transepithelial resistance [3, 10, 11]. More recent perspectives suggest, however, that sweat is a "constructed fluid" primarily under cholinergic control in humans; the process begins with active pumping of $Cl^-$ followed by $Na^+$ into the secretory coil to a high osmolality which in turn draws water into the lumen to re-equilibrate [2]. In this way, the preliminary sweat fluid produced by the eccrine secretory coil contains $Na^+$ and $Cl^-$ levels isotonic to plasma [3, 12]. This preliminary fluid is pumped along the duct where $Na^+$ and $Cl^-$ are then actively resorbed to varying degrees depending on the body location [3, 13].

Ductal resorption of $Na^+$ and $Cl^-$ has been attributed to the epithelial sodium channel (ENaC) and cystic fibrosis transmembrane regulator (CFTR) channel, occurring when $Na^+$ is pumped through the ENaC and $Cl^-$ follows passively. This is made possible as the loss of $Na^+$ from the lumen of the duct creates an electrochemical gradient allowing $Cl^-$ to enter the cell via the CFTR channels [14] which are ATP-dependent and pH sensitive [15]. Additional $Cl^-$ in the lumen is exchanged for $HCO_3^-$ via anion exchangers embedded in the luminal cell membrane [16]. The sweat released from the duct of the eccrine sweat glands contains $Na^+$ and $Cl^-$ ions at 10–30% of the concentration at which they occur in the plasma [17, 18]. In contrast, the level of $K^+$ in the final sweat is usually 2–3 times higher than found in plasma [3]. It has been suggested that secretion of $K^+$ into the sweat fluid within the duct may occur as a result of the active resorption of $Na^+$ but this is not fully understood [2]. Electrolyte resorption in the duct is important in avoiding exercise-associated hyponatremia (EAH), which occurs when the blood $[Na^+]$ falls below 135 mmol/L [19]. It commonly occurs in endurance athletes due to a combination of hypotonic fluid intake, reduced or no intake of $Na^+$ from food and losses of $Na^+$ from sweat and urine [20]. Concentrations below 120 mmol/L can result in fatigue, nausea, acute vomiting, acute epileptic seizures and neurological deficits [21].

Serine, histidine, alanine and ornithine are generally found at high levels in sweat compared with corresponding levels in plasma whilst glutamine and proline have been found at relatively low levels [4–7]. To date, little explanation for the high concentrations of amino acids in sweat has been proposed and there is no understanding of why glutamine and proline are selectively conserved. Shaping a detailed model of the cellular mechanisms that contribute to the high concentrations of amino acids in sweat is integral to determining the impacts of sweat facilitated losses of amino acids. Previous research has indicated that amino acid content in sweat decreased over the course of exercise [7]. An earlier study that collected and analysed sweat samples from the lower back of four athletes during exercise noted that $[Na^+]$ increased gradually for the first 40 minutes of exercise before plateauing at 26.2 mM (± 19.4 mM) [22]. The physiological impacts of increased $Na^+$ losses in sweat with time and diminished levels of amino acids may be linked with, and represent a mechanism for, heat adaptation.

Since the concentrations of certain amino acids and $K^+$ have been reported at significantly higher concentrations in sweat than in plasma [6, 23, 24], it was hypothesized that these components could be either directly or indirectly involved with the process of resorption of $Na^+$ and $Cl^-$ in the sweat duct. Alterations in the sweat concentrations of $Mg^{2+}$ and $Ca^{2+}$ could also potentially contribute to the resorption processes. It was thus proposed that if amino acids, $K^+$, $Mg^{2+}$ and $Ca^{2+}$ were involved in the resorption of $Na^+$ and $Cl^-$, then $[Na^+]$ and $[Cl^-]$ would be inversely correlated with concentrations of these sweat components over the duration of the exercise period. The aim of the current study was to investigate potential relationships between $[Na^+]$ and $[Cl^-]$ with $[K^+]$, $[Mg^{2+}]$, $[Ca^{2+}]$ and the amino acids in sweat collected from recreational athletes over a defined period of cycling exercise. The influence of exercise on sweat composition was also tested by comparisons with sweat collected under passive conditions of thermal stimulation.

## Materials and methods

### Ethics statement

All procedures performed in studies involving human participants were in accordance with the ethical standards of the institutional and/or national research committee and with the 1964 Helsinki declaration and its later amendments or comparable ethical standards.

Eleven recreational athletes were recruited from the University of Newcastle's Ourimbah and Callaghan campuses to participate in a 65-minute cycling session for sweat collection (seven males (weight 85 ±9 Kg, height 1.8 ± 0.05 m) and four females (weight 65 ± 7, height 1.7 ± 0.05 m), with an age of 28 ± 10y. The group undertook an average of 5.2 hours of exercise per week in the two weeks prior to the study (SD ±2.65 hours, range 2–10.5 hours). This research was approved by the University of Newcastle Human Research Ethics Committee (approval number: H-2014-0086) and the participants provided written informed consent prior to inclusion in the study.

All athletes provided a passive sweat sample from a pre-washed forearm by sitting in a heated tent area at 32˚C (±0.4˚C) for up to 15 minutes. All participants had their forearms washed with mild detergent and warm water up to the elbow, which was then dried with a clean paper towel. The collected sweat was aspirated into a Monovette™ and stored at 4˚C until analysed. However, only six of the eleven participants produced an adequate sweat sample during the passive sweat collection.

The participants underwent a YMCA Sub Maximal Cycle Ergometer test to determine maximum predicted workload [25, 26]. A plastic bag was sealed around one arm of each participant just below the elbow to collect sweat. The participants rode on a Monark cycle ergometer at 32˚C (±0.4˚C) and 42–57% relative humidity at 60% predicted $VO_2$ max for a total of 65 minutes. Sweat samples were collected following 15, 25, 35, 45, 55 and 65 minutes of exercise. The participant would dismount the cycle ergometer, the bag was removed, each participant's arm was re-washed and sealed with a clean bag. The sweat collected in the bag was transferred to a Monovette™ transport tube which was then stored at 4˚C until analysis.

Amino acids in the sweat samples were extracted and processed with a Phenomenex® EZ: faast™ sample kit for analyses by gas chromatography flame ionisation detection (GC-FID) as described previously [7]. A Zebron® 50% phenylpolysiloxane, 50% Methylpolysiloxane column (ZB-50) with dimensions 10 m x 0.25 mm and 0.25 μm film thickness, was used to run all samples. A pulsed splitless method was used with a starting temperature of 150˚C which increased at 32˚C per minute to a holding temperature of 320˚C. The flow rate was 2 mL/ minute and the carrier gas was Helium. The run time was 7.64 minutes in total. All GC-FID operations were controlled through Agilent® ChemStation™ software which had been calibrated

for each amino acid with standard concentrations of 5, 10, 20, 30 and 40 nm/100 μL against corresponding FID signal peak area. Norvaline (20 nm/100 μL) was added at the beginning of each sample preparation as the internal standard. The amino acids able to be integrated by this method included L-alanine, L-sarcosine, L-glycine, α-aminobutyric acid, L-valine, ß-aminoi-sobutyric acid, L-leucine, L-alloisoleucine, L-isoleucine, L-threonine, L-serine, L-proline, L-asparagine, L-thioproline, L-aspartic acid, L-methionine, L-hydroxyproline, L-glutamic acid, L-phenylalanine, α-aminoadipic acid, α-aminopimelic acid, L-glutamine, L-ornithine, glycyl-proline (dipeptide), L-lysine, L-histidine, L-hydroxylysine, L-tyrosine, proline-hydroxyproline (dipeptide), L-tryptophan, L-cystathionine and L-cystine. Total amino acid concentrations were based on summation of the measured individual amino acids.

To measure $Na^+$, $Mg^{2+}$, $K^+$ and $Ca^{2+}$ concentrations in sweat, samples were diluted 1:100 in 2% $NHO_3$ in a Labcon® MetalFree 15 mL tube and mixed thoroughly for 30 seconds. The samples were left to digest for 48–72 hours before 10 μL of an Agilent internal standard mix containing 100 ppm of Li6, Sc, Ge, In, Lu, Rh, Tb and Bi was added to each sample to produce a final concentration of 100 ppb. Each sample was run on an Agilent ICP-MS 7900. To deter-mine $Na^+$, $Mg^{2+}$, $K^+$ and $Ca^{2+}$ concentrations in each sample, High Purity Standards™ ICP-AM-15-1M mix was used to make calibration standards of concentrations 0, 1, 5, 10, 50, 100, 500, 1000, 5000 and 10000 ppb. These were prepared freshly and run immediately prior to each batch of sweat samples to produce a 10 point calibration curve employed upon batch completion by Agilent Technologies MassHunter 4.2 Workstation software to quantify $[Na^+]$. $[Cl^-]$ was determined in the same way with an 8 point calibration curve (0 ppb, 0.606 ppb, 6.06 ppb, 60.6 ppb, 0.606 ppm, 6.06 ppm, 15.15 ppm and 30.3 ppm) prepared from a 10,000 ppm stock solution of Fluka NaCl in Milli Q water. The internal standard mix was added to both calibration standards making a final concentration of 100 ppb.

Electrolyte and amino acid concentrations across the exercise protocol were analysed using one-way repeated measures ANOVAs. Upon significant time effects, further analysis were per-formed using Duncan's multiple range test. Additionally, Pearson's correlations were per-formed to assess interrelationships between outcome variables. All statistical analyses were conducted using TIBCO Software Inc. (2017), Statistica (data analysis software system), ver-sion 13 and statistical significance was accepted at $p < 0.05$.

## Results

The evaluations of electrolytes and amino acids in sweat have been summarised in Table 1 with reference to comparisons with literature values for plasma composition (electrolytes [27] and [28]; plasma [29]). $[Na^+]$ and $[Cl^-]$ were present in the passive sweat at 10% of the plasma concentrations, $[Mg^{2+}]$ at 22% and $[Ca^{2+}]$ at 34%, whereas $[K^+]$ was two times more concen-trated than in plasma. The average total level of amino acids was nearly four times higher in the passive sweat compared with the literature values for plasma. Serine, glycine, alanine, orni-thine and aspartic acid were the most abundant amino acid components in the sweat. Aspartic acid was 100 times more concentrated in the passive sweat than plasma and serine, ornithine and glycine were 26-fold, 14-fold and 7.4-fold higher than the corresponding literature plasma levels. Only trace levels of glutamine were detected in the passive sweat (Table 1).

The first sweat samples taken after 15 minutes of exercise showed marked differences in composition to the passive sweat samples (Table 1). The mean $[Na^+]$ and $[Cl^-]$ were increased by 8.9 mM and 6.25 mM respectively after 15 minutes of exercise compared with passive sweat, although these changes were not statistically significant. The $[K^+]$ was reduced by 2.1 mM and there were significant reductions in $[Mg^{2+}]$ and $[Ca^{2+}]$ of 0.08 and 0.29 mM respec-tively ($p < 0.05$). There was a significant reduction in the average total level of amino acids

**Table 1. The average concentrations of the electrolytes and amino acids in sweat collected from the recreational athletes were measured at six time points over the 65-minute period of exercise for comparison with literature plasma concentrations.** Passive sweat samples were collected from six participants.

| Electrolyte and amino acid concentrations (μM) | Passive sweat (n = 6) | Literature Plasma μM ± SE | Exercise sweat (n = 11) | | | | | | Correlations | Na⁺ | Cl⁻ | K⁺ | Mg²⁺ | Ca²⁺ |
|---|---|---|---|---|---|---|---|---|---|---|---|---|---|---|
| | | | 15 min | 25 min | 35 min | 45 min | 55 min | 65 min | | | | | | |
| Na⁺ | 14,385 ± 1226 | 140,400 ± 500 | 23,275 ± 3,053 | 28,703 ± 3,146 | 30,481 ± 2,950 | 30,488 ± 2,220 | 32,291 ± 2445 | 34,598 ± 2,366^ | Na⁺ | | 0.99 | -0.93 | -0.65 | -0.83 |
| Cl⁻ | 10,910 ± 1945 | 104,500 ± 500 | 17,163 ± 4,293 | 22,157 ± 3,922 | 25,335 ± 3,758 | 24,840 ± 2,031 | 25,573 ± 2789 | 28,595 ± 3,105 | Cl⁻ | | | -0.93 | -0.65 | -0.83 |
| K⁺ | 10,358 ± 685 | 4,420 ± 80 | 8,224 ± 929 | 6,250 ± 600 | 5,908 ± 639 | 5,517 ± 631 | 5,548 ± 625 | 5,489 ± 503 | K⁺ | | | | 0.82 | 0.96 |
| Mg²⁺ | 211 ± 26 | 964 ± 13 | 131 ± 24* | 91 ± 15 | 93 ± 14 | 98 ± 14 | 100 ± 12 | 104 ± 15 | Mg²⁺ | | | | | 0.95 |
| Ca²⁺ | 840 ± 95 | 2,483 ± 14 | 555 ± 75* | 324 ± 43 | 331 ± 44 | 313 ± 33 | 331 ± 33 | 333 ± 35^ | | | | | | |
| Serine | 3,030 ± 226 | 114 | 1,835 ± 336* | 1,454 ± 285 | 1,217 ± 267 | 1,241 ± 193 | 1,013 ± 186 | 1,004 ± 203 | Serine | -0.98 | -0.97 | 0.94 | 0.64 | 0.82 |
| Glycine | 1,751 ± 150 | 236 | 997 ± 202* | 804 ± 169 | 689 ± 154 | 700 ± 120 | 580 ± 99 | 595 ± 87 | Glycine | -0.97 | -0.96 | 0.94 | 0.66 | 0.83 |
| Alanine | 958 ± 96 | 419 | 601 ± 123 | 475 ± 102 | 426 ± 100 | 429 ± 79 | 342 ± 63 | 348 ± 53 | Alanine | -0.98 | -0.95 | 0.93 | 0.64 | 0.82 |
| Ornithine | 928 ± 144 | 65 | 452 ± 85* | 327 ± 60 | 272 ± 50 | 289 ± 43 | 273 ± 36 | 291 ± 40 | Ornithine | -0.89 | -0.90 | 0.97 | 0.86 | 0.95 |
| Aspartic acid | 702 ± 72 | 7 | 415 ± 99 | 299 ± 61 | 281 ± 60 | 277 ± 47 | 244 ± 41 | 245 ± 34 | Aspartic acid | -0.97 | -0.95 | 0.98 | 0.79 | 0.92 |
| Threonine | 661 ± 45 | 146 | 412 ± 77 | 319 ± 61 | 277 ± 60 | 283 ± 43 | 216 ± 34 | 211 ± 42 | Threonine | -0.98 | -0.96 | 0.92 | 0.62 | 0.80 |
| Histidine | 700 ± 73 | 89 | 260 ± 26 | 190 ± 30 | 159 ± 23 | 170 ± 15 | 134 ± 16 | 149 ± 19^ | Histidine | -0.95 | -0.93 | 0.96 | 0.75 | 0.88 |
| Valine | 433 ± 36 | 252 | 257 ± 48* | 208 ± 42 | 178 ± 39 | 187 ± 31 | 139 ± 29 | 152 ± 20 | Valine | -0.95 | -0.93 | 0.90 | 0.62 | 0.78 |
| Glutamic acid | 393 ± 47 | 60 | 168 ± 31* | 140 ± 27 | 122 ± 21 | 121 ± 12 | 95 ± 11 | 106 ± 14 | Glutamic acid | -0.94 | -0.92 | 0.91 | 0.60 | 0.78 |
| Leucine | 335 ± 35 | 160 | 162 ± 18* | 118 ± 13 | 102 ± 13 | 103 ± 8 | 95 ± 8 | 95 ± 8^ | Leucine | -0.95 | -0.96 | 0.99 | 0.80 | 0.93 |
| Proline | 252 ± 33 | 239 | 153 ± 30 | 114 ± 24 | 101 ± 22 | 106 ± 17 | 87 ± 14 | 89 ± 13 | Proline | -0.97 | -0.96 | 0.96 | 0.74 | 0.88 |
| Tyrosine | 292 ± 33 | 72 | 130 ± 20* | 93 ± 15 | 78 ± 14 | 77 ± 9 | 69 ± 8 | 69 ± 7^ | Tyrosine | -0.96 | -0.96 | 0.98 | 0.76 | 0.91 |
| Isoleucine | 224 ± 22 | 84 | 129 ± 19* | 94 ± 15 | 81 ± 14 | 83 ± 11 | 57 ± 12 | 67 ± 11^ | Isoleucine | -0.95 | -0.92 | 0.93 | 0.67 | 0.83 |
| Lysine | 265 ± 38 | 198 | 110 ± 20* | 82 ± 14 | 66 ± 13 | 67 ± 10 | 53 ± 8 | 65 ± 8^ | Lysine | -0.91 | -0.90 | 0.96 | 0.74 | 0.88 |
| Phenylalanine | 221 ± 25 | 65 | 95 ± 11* | 67 ± 8 | 56 ± 8 | 57 ± 5 | 52 ± 5 | 53 ± 5^ | Phenylalanine | -0.94 | -0.95 | 0.99 | 0.81 | 0.93 |
| Hydroxy proline | 93 ± 81 | | 82 ± 63 | 62 ± 50 | 47 ± 40 | 41 ± 33 | 8 ± 6 | 30 ± 29 | Hydroxy proline | -0.87 | -0.82 | 0.84 | 0.49 | 0.69 |
| Asparagine | 144 ± 15 | 49 | 79 ± 20* | 60 ± 14 | 47 ± 13 | 47 ± 13 | 40 ± 13 | 43 ± 8 | Asparagine | -0.94 | -0.94 | 0.96 | 0.70 | 0.86 |
| Tryptophan | 87 ± 15 | 30 | 38 ± 7* | 24 ± 6 | 18 ± 6 | 17 ± 4 | 13 ± 4 | 13 ± 3^ | Tryptophan | -0.97 | -0.96 | 0.98 | 0.74 | 0.90 |
| Glutamine | 4 ± 4 | 645 | 17 ± 9 | 13 ± 9 | 12 ± 9 | 5 ± 5 | 10 ± 6 | 9 ± 7 | Glutamine | -0.73 | -0.74 | 0.83 | 0.50 | 0.72 |
| Methionine | 43 ± 7 | 32 | 11 ± 3* | 6 ± 2 | 4 ± 3 | 4 ± 2 | 4 ± 3 | 6 ± 2^ | Methionine | -0.76 | -0.79 | 0.92 | 0.88 | 0.92 |
| Hydroxylysine | 8 ± 7 | | 8 ± 6 | 9 ± 8 | 5 ± 5 | 6 ± 6 | 3 ± 3 | 2 ± 2 | Hydroxylysine | -0.82 | -0.81 | 0.62 | 0.16 | 0.40 |
| α-aminoadipic acid | 14 ± 6 | | 6 ± 4 | 5 ± 4 | 5 ± 5 | 2 ± 2 | 2 ± 2 | 2 ± 2 | α-aminoadipic acid | -0.88 | -0.86 | 0.78 | 0.33 | 0.61 |
| α-aminobutyric acid | 5 ± 5 | | 5 ± 4 | 22 ± 19 | | | | | α-butyric acid | -0.33 | -0.39 | 0.23 | -0.23 | -0.03 |
| Total | 11,584 ± 831 | 2,962 | 6,422 ± 1187* | 4,975 ± 951 | 4,231 ± 870 | 4,305 ± 632 | 3,534 ± 541 | 3,639 ± 497 | Total | -0.97 | -0.96 | 0.95 | 0.69 | 0.85 |

The significant differences between the passive sweat and the sweat collected after 15 minutes of exercise have been marked * ($p<0.05$). Significant differences during the exercise period are marked ^ ($p<0.05$). Significant correlations between the mean values of Na⁺, Cl⁻, K⁺, Mg²⁺ and Ca²⁺ and the twenty-three most abundant amino acids over the six exercise time points have been marked in italics ($p<0.05$).

compared to the passive sweat (6.4 vs 11.6 mM) ($p<0.05$). Individual amino acids serine, glycine, ornithine, valine, glutamic acid, leucine, tyrosine, isoleucine, lysine, phenylalanine, asparagine, tryptophan and methionine were significantly lower in the exercise sweat at 15 minutes than in the passive sweat samples ($p<0.05$, Table 1).

Electrolyte concentrations changed over the duration of the exercise period. The [$Na^+$] and [$Cl^-$] increased from 23.3mM and 17.2mM at 15 minutes ($p<0.05$), to 34.6 mM and 28.6 mM respectively at 65 minutes (Fig 1(A)). In contrast, the total amino acid concentrations in sweat reduced from 6.4 mM at 15 minutes to 3.6 mM at 65 minutes although this was not significant. The [$K^+$] in exercise sweat declined from 8.2mM to 5.5mM at 65 minutes, but the [$Mg^{2+}$] remained relatively constant throughout the period of exercise (Fig 1(B)). [$Ca^{2+}$] decreased between 15 and 25 minutes but remained stable for the remaining 50 minutes of exercise.

As predicted, the [$Na^+$] and [$Cl^-$] in the exercise sweat were positively correlated throughout the exercise period ($r = 0.99$, $p<0.05$, Table 1). Neither [$Na^+$] nor [$Cl^-$] were positively associated with any other electrolyte or amino acid in the sweat but showed significant negative correlations with the total amino acid concentrations ($r = -0.97$, $r = -0.96$, $p<0.05$), $K^+$ ($r = -0.93$, $p<0.05$) and $Ca^{2+}$ ($r = -0.83$, $p<0.05$). Significant negative associations were demonstrated between [$Na^+$] and twenty of the individual amino acids measured, while [$Cl^-$] was negatively correlated with nineteen of the twenty three amino acids assessed during the exercise period (Table 1). The [$K^+$], [$Mg^{2+}$], [$Ca^{2+}$] in sweat were all positively correlated with each other ($r > 0.82$, $p<0.05$). [$K^+$] was positively correlated with twenty of the amino acids ($p<0.05$), [$Ca^{2+}$] was correlated with fifteen amino acids ($p<0.05$) and [$Mg^{2+}$] was associated with three amino acids ($p<0.05$).

## Discussion

The current study aimed to investigate potential relationships between [$Na^+$] and [$Cl^-$] with [$K^+$], [$Mg^{2+}$], [$Ca^{2+}$] and the amino acids in passive and exercise sweat collected from recreationally active participants over a defined period of cycling exercise. The results indicated that passive sweat collected over a 15 minute period had higher [$K^+$], [$Mg^{2+}$] and [$Ca^{2+}$] and concentrations of amino acids compared to sweat collected after 15 minutes of exercise. In contrast, the [$Na^+$] and [$Cl^-$] in the passive sweat were lower than those observed in the 15 minute exercise sweat. These differential profiles of amino acids and electrolytes in the passive and exercise sweats suggested that the composition of sweat was physiologically regulated by exercise. The advent of exercise would require the body to maintain sweating capacity to counter the higher heat generation for energy consumption in the muscles for prolonged periods and so a more conservative approach for retention of $Na^+$ and $Cl^-$ would be required. Under these conditions of exercise, the body would limit the rate of losses of the amino acids and $K^+$, $Ca^{2+}$ and $Mg^{2+}$ and, as a result, the capacity to resorb the $Na^+$ and $Cl^-$ would be reduced, resulting in higher [$Na^+$] and [$Cl^-$] in exercise sweat. It should be noted that $K^+$ was the only ion in the exercise sweat at a concentration (8.2mM) greater than the literature average value in plasma (4.4mM). This would be consistent with $K^+$ having a major role in the process of sweat formation and/or resorption of $Na^+$.

During the exercise period, it was observed that there were progressive increases in in [$Na^+$] and [$Cl^-$] in the sweat, whilst the [$K^+$], [$Mg^{2+}$] and [$Ca^{2+}$] and amino acids diminished. In the current study, the exercise intensity and environmental conditions were kept constant to enable a constant sweat rate [6, 30]. The progressive increases in [$Na^+$] and [$Cl^-$] were thus interpreted to indicate that the capacity to resorb $Na^+$ and $Cl^-$ during passage through the duct diminished over the exercise period. In contrast, the [$K^+$] in the sweat fell after 15 minutes of exercise and continued to decrease over time to levels that were closer to, but still higher than,

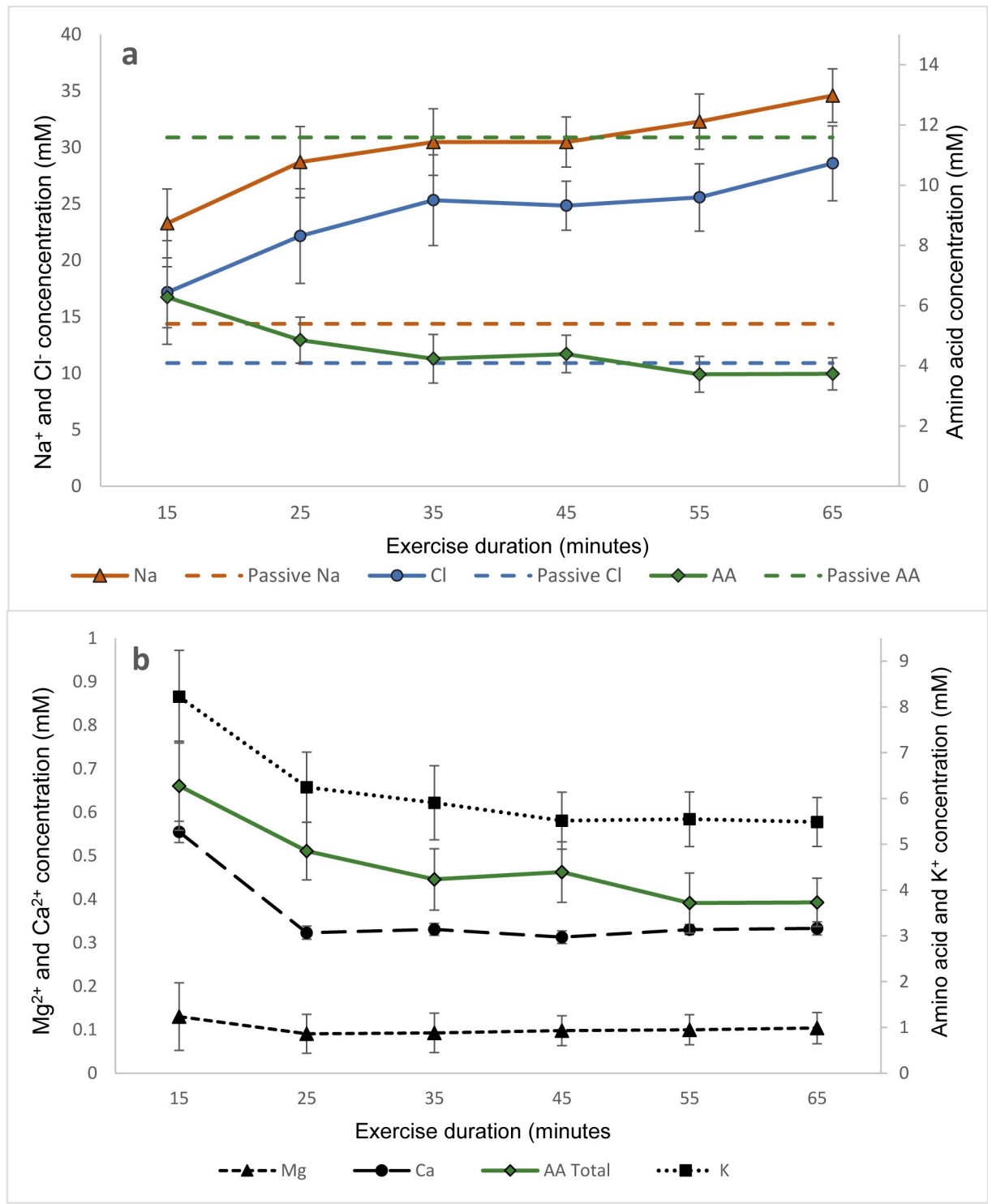

**Fig 1.** (a) Concentrations of Na$^+$ and Cl$^-$ (primary axis) and mean total amino acid concentrations (secondary axis) in exercise sweat (mM) assessed at ten minute intervals from 15 to 65 minutes of exercise (n = 11). The concentrations of Na$^+$, Cl$^-$ and total amino acids in passive sweat (n = 6) are also shown for reference on the respective axes. (**b**), Mg$^{2+}$ and Ca$^{2+}$ concentrations (primary axis) in exercise sweat for the exercise duration are shown with amino acid and K$^+$ concentrations (secondary axis).

the literature plasma levels by 65 minutes. The mean total level of amino acids was also approximately twice the literature plasma value after 15 minutes exercise and progressively diminished over time approaching levels equivalent to plasma by 55–65 minutes. As a result, there were strong positive correlations between the sweat concentrations of amino acids and $[K^+]$, with both these components having strong negative correlations with $[Na^+]$ and $[Cl^-]$. These results suggested inverse physiological relationships between $[Na^+]$ and $[Cl^-]$ in the sweat with $[K^+]$ and the amino acids which were modulated by the onset and duration of exercise.

The *in vivo* amino acid content of the primary sweat fluid produced in the secretory coil has not been examined independently of sweat leaving the resorptive duct, as was achieved with *in vitro* studies of sweat electrolytes from excised glands [31]. It would be reasonable to assume that the amino acid levels in the primary sweat fluid formed in the secretory coil would be less than or similar to plasma levels with no published literature evidence of amino acid transporters operating in this part of the gland or throughout the transdermal duct. A potential transporter that facilitates the exchange of amino acids for extracellular substrates is the ASCT-1 protein of the alanine/serine/cysteine transporter family (ASCT) [32]. The expression of ASCT-1 mRNA was found to be expressed in all the human tissue types that were tested by northern blot, with the highest abundances occurring in the brain, muscle and pancreatic tissue [33]. The ASCT-1 protein transports alanine, serine, cysteine and threonine in a symmetrical and electroneutral exchange of one of these zwitterionic amino acids with one $Na^+$ ion [34]. As the ASCT-1 has a high affinity for serine and alanine, the presence of this transporter in the duct of the sweat gland may provide a supplementary mechanism by which $Na^+$ is reclaimed at the cost of small, neutral amino acids. The operation of this transporter would be consistent with the very high levels of serine and alanine measured in the sweat. Other amino acid transport systems are likely to be involved given the strong inverse correlations between $[Na^+]$ and $[Cl^-]$ with the concentrations of eighteen amino acids, $[K^+]$, $[Mg^{2+}]$ and $[Ca^{2+}]$ in the sweat.

Once the primary sweat fluid has been formed in the secretory coil, the amino acids and $K^+$ could be transferred to the primary sweat fluid during passage through the intradermal duct to balance the resorption of $Na^+$ and $Cl^-$. It has been proposed that $K^+$ is lost from the duct cells via leak channels as a consequence of $Na^+$ resorption in an attempt to maintain osmotic balance in the cells but not necessarily as a direct part of the $Na^+$ transport channel operation [18, 31]. In this context, $Na^+$ would be resorbed from the primary sweat fluid into the cells via the ENaC channels as it travels along the duct. The change in the ion concentration gradient causes $K^+$ to exit the cell passively through $K^+$ leakage channels, resulting in elevated $K^+$ concentrations in the sweat at the skin surface. The amino acids may be linked to this process or have independent mechanisms of transfer. $Ca^{2+}$ and $Mg^{2+}$were also implicated in the resorption process by showing positive correlations with $K^+$ and the amino acids. Whilst $Ca^{2+}$ has been linked to KCl efflux channels [24, 35, 36], the full role and mode of action of $Mg^{2+}$and $Ca^{2+}$ has not been clearly elucidated and the connection between the cations and amino acids is at present undefined.

The increases in $[Na^+]$ and $[Cl^-]$ throughout the exercise period indicated that the capacity for resorption diminished over this period. It was thus proposed that whilst sufficient amino acids and $K^+$ were available, the $Na^+$ can be effectively resorbed but as exercise and sweating continues and reserves of amino acids and $K^+$ diminish, then $Na^+$ (and $Cl^-$) would not be as effectively resorbed and thus $Na^+$ and $Cl^-$ would begin to increase in concentration within the sweat matrix. If the concentrations of amino acids and $K^+$ became limiting within the duct cells with exercise duration, then fewer $Na^+$ ions could be resorbed progressively resulting in higher $Na^+$ and lower $K^+$ concentrations in sweat as was observed in the present study.

## Conclusions

The data from this study supported the hypothesis that key amino acids in combination with $K^+$, $Mg^{2+}$ and $Ca^{2+}$, are transferred into the sweat to facilitate resorption of $Na^+$ and $Cl^-$. The involvement of the amino acids, $K^+$, $Mg^{2+}$ and $Ca^{2+}$ in the resorption of $Na^+$ and $Cl^-$ would provide an explanation for the high losses of these components via sweat during exertion. During physical activity, reclamation of $Na^+$ and $Cl^-$ is important in avoiding exercise-associated hyponatremia (EAH). The implication for professional and recreational athletes is that provision of electrolytes alone would not be sufficient to balance the replenishment requirements generated by prolonged exertion. These results could provide new insights in sports medicine to manage athletes undertaking high intensity and endurance training or competition, especially in warmer climates. It would even be possible to use measures of sweat-facilitated losses of amino acids to customise supplementation to minimize the supply from endogenous turnover of body proteins. Ideally, the amino acids and electrolytes would be replaced in a similar concentration profile as to that in which they are lost, if possible during, or immediately after exercise.

## Author Contributions

**Conceptualization:** R. Hugh Dunstan, Margaret M. Macdonald, Nattai Borges, Zoe Radford, Diane L. Sparkes, Benjamin J. Dascombe, Timothy K. Roberts.

**Formal analysis:** Grace R. Murphy, R. Hugh Dunstan, Nattai Borges, Zoe Radford, Diane L. Sparkes.

**Funding acquisition:** R. Hugh Dunstan.

**Investigation:** Grace R. Murphy, R. Hugh Dunstan, Nattai Borges, Zoe Radford, Diane L. Sparkes, Benjamin J. Dascombe, Timothy K. Roberts.

**Methodology:** Grace R. Murphy, R. Hugh Dunstan, Margaret M. Macdonald, Nattai Borges, Zoe Radford, Diane L. Sparkes, Benjamin J. Dascombe, Timothy K. Roberts.

**Project administration:** R. Hugh Dunstan, Margaret M. Macdonald, Diane L. Sparkes.

**Resources:** R. Hugh Dunstan, Nattai Borges, Benjamin J. Dascombe, Timothy K. Roberts.

**Supervision:** R. Hugh Dunstan, Margaret M. Macdonald, Nattai Borges, Benjamin J. Dascombe.

**Validation:** Grace R. Murphy, R. Hugh Dunstan, Margaret M. Macdonald, Zoe Radford, Diane L. Sparkes.

**Writing – original draft:** Grace R. Murphy, R. Hugh Dunstan.

**Writing – review & editing:** Grace R. Murphy, R. Hugh Dunstan, Margaret M. Macdonald, Nattai Borges, Zoe Radford, Diane L. Sparkes, Benjamin J. Dascombe, Timothy K. Roberts.

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
