## [Decision Letter · Decision Letter 0]

12 Sep 2019

PONE-D-19-20805

Relationships between electrolyte and amino acid compositions in sweat during exercise suggest a role for amino acids and K+ in reabsorption of Na+ and Cl- from sweat

PLOS ONE

Dear Dr R. Hugh Dunstan,

Thank you for submitting your manuscript to PLOS ONE. After careful consideration, we feel that it has merit but does not fully meet PLOS ONE’s publication criteria as it currently stands. Therefore, we invite you to submit a revised version of the manuscript that addresses the points raised during the review process.

ACADEMIC EDITOR:

I suggest to

a) include all the analytical figures of merit of the procedures used to analyse sweat samples

b) conclusions, underline the possible applications of these results in the field of the sport medicine.

c) introduction, explain the reasons of selecting sweat samples instead of other non-conventional biological fluids (e.g. saliva). Saliva analysis is a well-know approach to obtain information of the physiological status in both clinical and sport medicine field. As example, the following articles can be useful for the authors and used in the introduction.

DOI: 10.1016/j.microc.2017.04.033

DOI: 10.1016/j.microc.2017.02.010

DOI: 10.1016/j.aca.2017.07.050

Sports Med. 1998 Jul;26(1):17-27.

We would appreciate receiving your revised manuscript by 29th September. To enhance the reproducibility of your results, we recommend that if applicable you deposit your laboratory protocols in protocols.io, where a protocol can be assigned its own identifier (DOI) such that it can be cited independently in the future. For instructions see: http://journals.plos.org/plosone/s/submission-guidelines#loc-laboratory-protocols

We look forward to receiving your revised manuscript.

Kind regards,

Tommaso Lomonaco, Ph.D

Academic Editor

PLOS ONE

Reviewers' comments:

Reviewer's Responses to Questions

**Comments to the Author**

1. Is the manuscript technically sound, and do the data support the conclusions?

Reviewer #1: Yes

2. Has the statistical analysis been performed appropriately and rigorously? 

Reviewer #1: Yes

3. Have the authors made all data underlying the findings in their manuscript fully available?

Reviewer #1: Yes

4. Is the manuscript presented in an intelligible fashion and written in standard English?

Reviewer #1: Yes

5. Review Comments to the Author

Reviewer #1: In this work the authors' purpose was to evaluate the potential relationships between [Na+] and [Cl-] with [K+], [Mg2+], [Ca2+] and the amino acids in sweat, and the role of exercise on sweat composition. The authors showed that the exercise stimulates a change in sweat composition with an increase of [Na+] and decrease of amino acids concentration. The results suggest that amino acids, [K+], [Ca2+] are associated with reabsorption of [Na+] and [Cl-].

The work turns out to be well written, the sources cited are appropriate.

The idea turns out to be innovative and the results are very interesting. Indeed, the results will lay important bases on understanding biochemistry of physical exercise and physiological interactions. Not only that, they will also allow to better understand the type of supplement to be carried out during a training and if it is the case to carry it out.

I have only few questions:

a) The authors did not carry out the assessment of body composition in terms of fat and lean mass.

Do you think that there could be an influence in the results in relation to the change in fat and lean mass percentages?

b) The exercise in the study is of an aerobic nature.

According to by Kerksick et al. (2018) [Kerksick CM, Wilborn CD, Roberts MD, et al. ISSN exercise & sports nutrition review update: research & recommendations. J Int Soc Sports Nutr. 2018 Aug 1;15(1):38. doi: 10.1186/s12970-018-0242-y], the position of the International Society of Sports Nutrition (ISSN) in order to achieve an improvement in performance, the only amino acid (AA) that seems to play a role is beta-alanine, while for the other AA the results they are contradictory.

Because on their results, do the authors believe that an integration with essential amino acids (EAAs) and branched amino acids (BCAAs) could improve performance? If yes, when would it be fairer to hire them? Before, during or after the performance?

6. PLOS authors have the option to publish the peer review history of their article (what does this mean?). If published, this will include your full peer review and any attached files.

Reviewer #1: No

---

## [Author Response · Author response to Decision Letter 0]

19 Sep 2019

This is more readily viewed in the response to reviewers file submitted as an attachment.

Response to Reviewers:

ACADEMIC EDITOR:

I suggest to

a) include all the analytical figures of merit of the procedures used to analyse sweat samples

This method has been described previously in our earlier PLOS publication on human sweat. We should have referenced this before. This has been added in lines 128 – 9. 

b) conclusions, underline the possible applications of these results in the field of the sport medicine.

The conclusions have been amended as requested.

c) introduction, explain the reasons of selecting sweat samples instead of other non-conventional biological fluids (e.g. saliva). Saliva analysis is a well-know approach to obtain information of the physiological status in both clinical and sport medicine field. As example, the following articles can be useful for the authors and used in the introduction.

DOI: 10.1016/j.microc.2017.04.033

DOI: 10.1016/j.microc.2017.02.010

DOI: 10.1016/j.aca.2017.07.050

Sports Med. 1998 Jul;26(1):17-27.

The intent of the study was not to try to determine physiological status of amino acids in the body. We specifically set out to investigate the nature of amino acid losses in sweat during exercise and how this might be related to losses of electrolytes. Sweat losses can occur at rates of 1-2 L per hour depending on the exercise intensity, type and external conditions. Thus losses of electrolytes and amino acids can be considerable.

In the introduction, it is pointed out that sweat contains certain amino acids at very high concentrations (line 44… and again at line 82) – many times higher than in plasma, but little is known as to why they are there at high levels – what function are they serving?

The perspective of sweat as a constructed fluid is then presented which is well understood in terms of electrolyte composition and re-absorption of Na and Cl, but there is nothing published about how and why the amino acids actually enter the sweat fluid.

The hypothesis is thus proposed that the amino acids and K may have direct roles in Na resorption. The project thus focusses on gathering data that might support this hypothesis and shed light on mechanisms.

We would appreciate receiving your revised manuscript by 29th September. To enhance the reproducibility of your results, we recommend that if applicable you deposit your laboratory protocols in protocols.io, where a protocol can be assigned its own identifier (DOI) such that it can be cited independently in the future. For instructions see: http://journals.plos.org/plosone/s/submission-guidelines#loc-laboratory-protocols

• A rebuttal letter that responds to each point raised by the academic editor and reviewer(s). This letter should be uploaded as separate file and labeled 'Response to Reviewers'.

• A marked-up copy of your manuscript that highlights changes made to the original version. This file should be uploaded as separate file and labeled 'Revised Manuscript with Track Changes'.

• An unmarked version of your revised paper without tracked changes. This file should be uploaded as separate file and labeled 'Manuscript'.

We look forward to receiving your revised manuscript.

Kind regards,

Tommaso Lomonaco, Ph.D

Academic Editor

PLOS ONE

We have altered the referencing format as requested. 

We have changed the headings style as requested.

We have changed the author naming as requested and amended the information for the corresponding author. We have also marked the “equally contributing” authors.

Reviewers' comments:

Reviewer's Responses to Questions

Comments to the Author

1. Is the manuscript technically sound, and do the data support the conclusions?

Reviewer #1: Yes

2. Has the statistical analysis been performed appropriately and rigorously? 

Reviewer #1: Yes

3. Have the authors made all data underlying the findings in their manuscript fully available?

Reviewer #1: Yes

4. Is the manuscript presented in an intelligible fashion and written in standard English?

Reviewer #1: Yes

5. Review Comments to the Author

Reviewer #1: In this work the authors' purpose was to evaluate the potential relationships between [Na+] and [Cl-] with [K+], [Mg2+], [Ca2+] and the amino acids in sweat, and the role of exercise on sweat composition. The authors showed that the exercise stimulates a change in sweat composition with an increase of [Na+] and decrease of amino acids concentration. The results suggest that amino acids, [K+], [Ca2+] are associated with reabsorption of [Na+] and [Cl-].

The work turns out to be well written, the sources cited are appropriate.

The idea turns out to be innovative and the results are very interesting. Indeed, the results will lay important bases on understanding biochemistry of physical exercise and physiological interactions. Not only that, they will also allow to better understand the type of supplement to be carried out during a training and if it is the case to carry it out.

I have only few questions:

a) The authors did not carry out the assessment of body composition in terms of fat and lean mass.

Do you think that there could be an influence in the results in relation to the change in fat and lean mass percentages?

This could be relevant. Eccrine glands do not have “fatty” secretions like the apocrine glands but there could be a link between the body types and sweat composition. Our first study indicated that concentrations of amino acids could vary between individuals, which may be related to various body types and fat composition. Future work could certainly target comparisons of sweat content with fat and lean mass. 

b) The exercise in the study is of an aerobic nature.

According to by Kerksick et al. (2018) [Kerksick CM, Wilborn CD, Roberts MD, et al. ISSN exercise & sports nutrition review update: research & recommendations. J Int Soc Sports Nutr. 2018 Aug 1;15(1):38. doi: 10.1186/s12970-018-0242-y], the position of the International Society of Sports Nutrition (ISSN) in order to achieve an improvement in performance, the only amino acid (AA) that seems to play a role is beta-alanine, while for the other AA the results they are contradictory.

Because on their results, do the authors believe that an integration with essential amino acids (EAAs) and branched amino acids (BCAAs) could improve performance?

Our research has also focussed on modelling amino acid turnover with specific reference to including exact losses via excretory pathways of sweat and urine. This modelling shows that a select few amino acids are subject to short supply as exercise demand increases: either as essential amino acids or conditionally essential amino acids. During exercise, some amino acids are utilised at faster rates than others, and these are supplied endogenously by catabolism of muscle proteins. See our ref below:

Dunstan, R. H., Macdonald, M. M., Murphy, G. R., Thorn, B., & Roberts, T. K. (2019). Modelling of protein turnover provides insight for metabolic demands on those specific amino acids utilised at disproportionately faster rates than other amino acids. Amino Acids. doi:10.1007/s00726-019-02734-1

We did not reference this article in this manuscript because here, the focus was on the associations between amino acids and electrolytes during the course of exercise.

The modelling paper basically infers that if you can provide the amino acids that are lost at disproportionately faster rates than others, then you could reduce the need for endogenous turnover of proteins. This would minimise wastage of amino acids that are not necessarily required in high demand – such as the branch chain amino acids.

 If yes, when would it be fairer to hire (=take) them?

It would be proposed to take an amino acid supplement that was designed to replenish those most rapidly lost during the exercise via sweat. Since digestion is impaired during exercise, it is possible to provide the free amino acids that can be directly absorbed to replenish those lost at the faster rates.

 Before, during or after the performance?

All three stages are possible. It should be noted that digestion can be impaired during and for several hours after the exercise, dependent on the intensity and duration. However, the repair and recovery processes are underway and continue after cessation of exercise, even if digestion is minimal. So it would be recommended to take amino acid supplementation as soon as possible after exercise. It is also proposed that taking a combination mix of electrolytes and amino acids during exercise would be advantageous.

6. PLOS authors have the option to publish the peer review history of their article (what does this mean?). If published, this will include your full peer review and any attached files.

Do you want your identity to be public for this peer review? For information about this choice, including consent withdrawal, please see our Privacy Policy.

Reviewer #1: No

---

## [Editor Report · Decision Letter 1]

20 Sep 2019

Relationships between electrolyte and amino acid compositions in sweat during exercise suggest a role for amino acids and K+ in reabsorption of Na+ and Cl- from sweat

PONE-D-19-20805R1

Dear Dr. Dunstan,

We are pleased to inform you that your manuscript has been judged scientifically suitable for publication and will be formally accepted for publication once it complies with all outstanding technical requirements.

With kind regards,

Tommaso Lomonaco, Ph.D

Academic Editor

PLOS ONE

Additional Editor Comments (optional):

Dear Authors,

the present version of the article is fine and can be accepted in PlosOne for publication.

Best regards,

Tommaso Lomonaco